# Candida albicans Adhesion Measured by Optical Nanomotion Detection

**Maria I. Villalba [1,2], Salomé LeibundGut-Landmann [3], Marie-Elisabeth Bougnoux [4,5], Christophe d'Enfert [4], Ronnie G. Willaert [2,6] and Sandor Kasas [1,2,7,*]**

1   Laboratory of Biological Electron Microscopy, Swiss Federal Institute of Technology Lausanne (École Polytechnique Fédérale de Lausanne), 1015 Lausanne, Switzerland; ines.villalba@epfl.ch

2   International Joint Research Group VUB-EPFL NanoBiotechnology & NanoMedicine (NANO), Vrije Universiteit Brussel, 1050 Brussels, Belgium; Swiss Federal Institute of Technology Lausanne (Ecole Polytechnique Fédérale de Lausanne), 1015 Lausanne, Switzerland; ronnie.willaert@vub.be

3   Section of Immunology, Vetsuisse Faculty, and Institute of Experimental Immunology, University of Zurich, 8006 Zurich, Switzerland; salome.leibundgut-landmann@uzh.ch

4   Unité Biologie et Pathogénicité Fongiques, Institut Pasteur, Université Paris Cité, INRAE USC 2019, 75015 Paris, France; marie-elisabeth.bougnoux@aphp.fr (M.-E.B.); christophe.denfert@pasteur.fr (C.d.)

5   Assistance Publique des Hôpitaux de Paris (APHP), Unité de Parasitologie-Mycologie, Service de Microbiologie Clinique, Hôpital Necker-Enfants-Malades, 75015 Paris, France

6   Research Group Structural Biology Brussels (SBB), Alliance Research Group VUB-UGent NanoMicrobiology (NAMI), Vrije Universiteit Brussel, 1050 Brussels, Belgium

7   Centre Universitaire Romand de Médecine Légale, UFAM, Université de Lausanne, 1015 Lausanne, Switzerland

*   Correspondence: sandor.kasas@epfl.ch

**Abstract:** Cellular adhesion plays an important role in numerous fundamental physiological and pathological processes. Its measurement is relatively complex, requires sophisticated equipment, and, in most cases, cannot be carried out without breaking the links between the studied cell and its target. In this contribution, we propose a novel, nanomotion-based, technique that overcomes these drawbacks. The applied force is generated by the studied cell itself (nanomotion), whereas cellular movements are detected by traditional optical microscopy and dedicated software. The measurement is non-destructive, single-cell sensitive, and permits following the evolution of the adhesion as a function of time. We applied the technique on different strains of the fungal pathogen *Candida albicans* on a fibronectin-coated surface. We demonstrated that this novel approach can significantly simplify, accelerate, and make more affordable living cells–substrate adhesion measurements.

**Keywords:** *Candida albicans*; cellular nanomotion; optical nanomotion detection; yeast adhesion; fibronectin





## 1. Introduction

Adhesion is a fundamental property of living cells that permits their attachment to other organisms or various organic and inorganic substrates. It plays a fundamental role in numerous physiological and pathological processes such as cell growth, migration, immune response, pathogen-host interaction, and tumor cell growth and spreading [1–5]. Despite the existence of different techniques to measure adhesion, its quantification is still a challenge [6–9]. A comprehensive review of cellular adhesion's importance in physiological and pathological processes, as well as measurement techniques, can be found in [10]. Most of the existing techniques rely on the detachment of the cells upon an applied force. Liquid flow (shear stress) [11–14], centrifugal acceleration [15,16], micropipette manipulation [17,18], optical tweezers [19,20], atomic force microscopy [6,21,22] or FRET force sensors [23] can generate this force. However, applying it with a force in the range of nano-newton on single cells is not a trivial task, and, therefore, the setup of such measurements is relatively complex and relies on expensive equipment. In addition,

detachment-based measurements do not permit obtaining information about the adhesion process as a function of time at a single-cell level. An alternative to the force-applying devices is the use of planar optical waveguides that monitor the contact surface between the living organism and the substrate [24]. This last approach requires specially treated surfaces and only informs about the surface of contact between the cell and its substrate.

In recent years, it was highlighted that all living organisms oscillate at a nanometric scale as long as they are alive. These oscillations, referred to as nanomotion, since the displacements are in the nanometer range, stop as soon as the organism is dead. Nanomotion exists in virtually all living organisms on earth [25–30]. Its monitoring is nowadays applied to rapid antibiotic sensitivity tests; for example, if a bacterium is sensitive to a given drug, its nanomotion rapidly stops or decreases upon antibiotic exposure [30–33]. Nanomotion was initially highlighted by atomic force microscopy (AFM), but later it appeared that classical optical microscopes equipped with a camera also detect nanomotion [34]. The technique consists of recording a movie of the living organism and processing it with motion detection dedicated software that allows tracking bacterial or fungal displacements with a sub-pixel resolution; therefore, the method is referred to as optical nanomotion detection (ONMD). This last development dramatically increased the availability of the technique and its accessibility in terms of cost and complexity of use. The origin of cellular nanomotion is still under debate; the opening and closing of ion channels, rearrangements of the cytoskeleton, or conformational changes of membrane proteins certainly participate in its generation [35].

In the present study, we introduced a novel nanomotion-based method to measure single-cell adhesion of the human fungal pathogen *Candida albicans* on flat surfaces. The technique relies on the detection of cellular nanomotion, and is therefore essentially limited to living organisms. We focused in this manuscript to the adhesion to fibronectin since it is a physiologically relevant molecule that is present in patients' basal lamina [36] and plays a role in *Candida* adhesion in immunocompromised patients. It has been shown that fibronectin is an important protein ligand of the host extracellular matrix (ECM) that plays an essential role in *C. albicans* adhesion [37–40]. Furthermore, targeting fibronectin has shown to alter *C. albicans* biofilm formation [41]. The nanomotion-based adhesion measurement relied on the position of the studied cell (a *C. albicans* cell, in this study) as a function of time. Depending on the attachment force and the length of the link between the cell and the substrate, the displacements of the cell will be more or less constrained, i.e., the displacement envelope of a cell is inversely proportional to the adhesive force and the length of the anchoring linker. A strongly attached organism will be constrained to a smaller area than a loosely attached one. To confirm this hypothesis, we examined the nanomotion amplitude of different *C. albicans* strains deposited on an optical quality Petri dish coated with fibronectin. We measured the amplitude of nanomotion of the different strains and compared it to the values obtained by classical adhesion tests. It appeared that our hypothesis was confirmed, and strongly attached cells had a lower nanomotion amplitude than those that were loosely attached.

## 2. Materials and Methods

### 2.1. Yeast Cell Culture

In this study, *C. albicans* strains CEC3672 [42], CEC3609 [43], CEC3678 [42], CEC3621 [44], SC5314 [45], 101 [46] and CEC3675 [42] were used. The yeasts were plated from cryo stocks on yeast-extract peptone–dextrose (YPD) agar and cultured for 24 h at 30 °C. YPD was composed of d-glucose (Gibco™ 15023021, Grand Island, NY, USA) 20 g/L, peptone (Millipore 82303, Burlington, MA, USA) 20 g/L, yeast extract 10 g/L, and agar (plates only) 20 g/L, which was diluted in distilled water. Single colonies from the agar plates were then grown in 4 mL YPD liquid medium at 30 °C, shaking at 160 rpm overnight.

### 2.2. Adhesion Assay

Two different methods were used to measure *C. albicans* adhesion, as illustrated in Figure 1. The first method [46–48] consisted of incubating freely floating yeast cells in wells coated with fibronectin (Figure 1A). A 6-well culture plate (Thermo Scientific™-Nunc™, 140675, Waltham, MA, USA) was incubated for an hour with fibronectin (50 µg/mL) (fibronectin human plasma, lyophilized powder, Sigma-Aldrich, F2006, St. Louis, MO, USA). A PBS wash was performed following the incubation with fibronectin. The liquid was gently removed, and 250 cells suspended in 500 µL of YPD medium were added to each well. After an incubation at 30 °C without shaking during 25 min, the supernatant containing the freely floating cells was removed and inoculated in a new YPD agar Petri dish (Figure 1B); attached cells were immersed in Sabourand dextrose (SDA) agar, (Figure 1C), by adding melted agar at 40 °C. SDA was composed of d-glucose (Gibco™ 15023021, Grand Island, NY, USA) 40 g/L, peptone (Millipore 82303, Burlington, MA, USA) 10 g/L, and agar 15 g/L, which was diluted in distilled water. After an incubation of 48 h the yeast cells in both Petri dishes formed visible colonies. The % adhesion was calculated by dividing the number of grown cells in SDA agar by the total number of grown cells (in both YPD and SDA agar) for each strain.

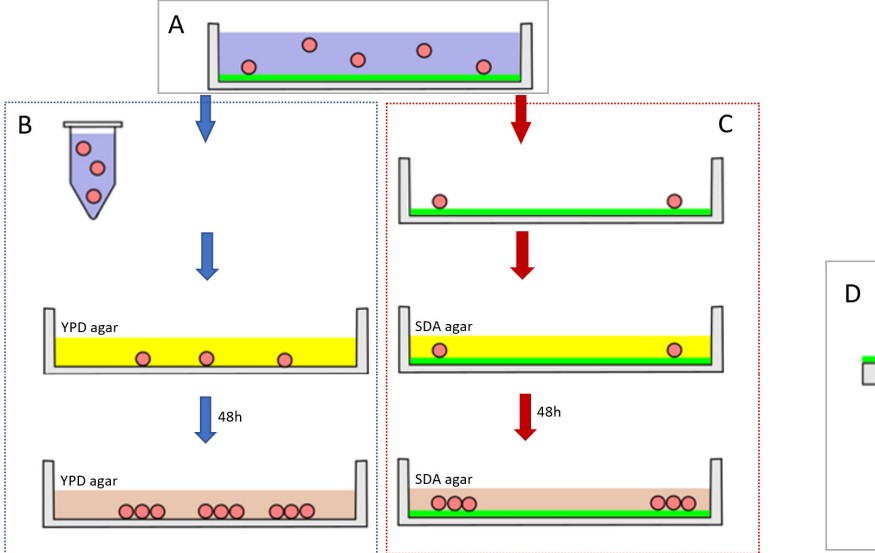
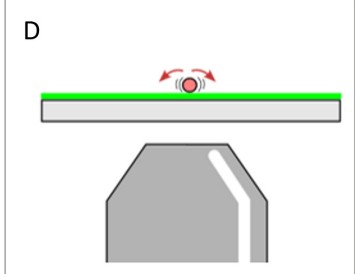

**Figure 1.** Two different techniques are used to measure adhesion: (**A**–**C**) Classical technique based on the ratio between floating and adhering cells and (**D**) nanomotion-based measurement. The classical method (**A**–**C**) consists of depositing yeast cells onto a fibronectin-coat Petri dish (**A**). The supernatant is than removed (**B**), then deposited into a new Petri dish with YPD agar. The single cells eventually divide and grow into colonies that enable the estimation their number. The cells that remained attached to the fibronectin covered Petri dish (**C**) were also covered with SDA agar, and developed CFUs, too. The nanomotion-based measurements (**D**) were performed by recording movies of 10 s.

### 2.3. Optical Nanomotion Adhesion Method

ONMD was performed by recording 10 s long movies with an inverted optical microscope (Zeiss Observer Z.1) of the attachment of *C. albicans* cells (Figure 1D). We used a 63× oil immersion objective, and a PCO Edge 5.5 camera. Petri dishes (µ-Dish 35 mm, low, uncoated, 80131 IBIDI) were incubated for an hour with fibronectin (50 µg/mL) (Sigma-Aldrich, F2006, St. Louis, MO, USA), followed by washing with PBS, according to the protocol recommended at www.ibidi.com. The liquid was removed, and 1.5 mL of the YPD liquid medium with 20 µL of overnight culture was added to the Petri dish with the fibronectin layer. Two minutes after placing the plates under the microscope, the first video (time 0) was recorded, followed by videos at 5, 10, 20, 30, and 40 min. All the measurements were carried out at room temperature without phase contrast nor fluorescent staining.

The camera typically recorded 10 s long videos (AVI format) at a frame rate of 30 frames per second (fps). The movies were processed with a custom-made tracking algorithm [34] implemented in Matlab, which tracked individual cells in 2D along 100–200 frames. Only individual cells (those that were not in contact with other cells) were selected. The algorithm reached a sub-pixel resolution. The Matlab program is freely available upon request. The total displacements of the cells were calculated and the average value represented the optical nanomotion signal [34].

### 2.4. Constrained Random Walk Simulation

A fixed step length random walk [49] was simulated with a Matlab (R2023a) computer program. We simulated a massless particle submitted to a random walk during 50,000 time cycles. We carried out two simulation rounds: one with a "rope" of 80 arbitrary length units, which restrains the distance to which the particle can diffuse, and a second round that simulates a free, "ropeless" particle diffusion. The random walk simulation consisted of randomly choosing a displacement direction at every simulation step and displacing the particle in that direction with an arbitrary unit length. In the "particle attached to a rope" simulation, the software checked the distance between the spot from which the simulation started (in our case $x = 0$ and $y = 0$) and the particle's actual position. If it exceeds that of the rope, another random displacement direction is tested and "refused", unless it decreases the distance between the particle and its starting position.

### 2.5. Statistical Analysis

Statistical analysis was performed using OriginPro, version 2021. We used the *t*-test to analyze the significance of the results (* $p < 0.05$). Each experiment was replicated at least three times. Optical nanomotion was analyzed for each case using more than 30 individual cells. For the adhesion assay, more than eight replicates were conducted.

## 3. Results

To confirm our hypothesis, according to which a low adhesion force corresponds to a higher displacement freedom, we carried out constrained and non-constrained random walk simulations. If assuming that the grid lines of the graphs delimit single pixels, Figure 2 clearly shows that a constrained particle "visits" (gray color squares) much less pixels than a free one. In the present simulation, only four pixels were visited by the attached particle, whereas 13 were visited by the free one.

We then compared seven different *C. albicans* strains by both the classical cell attachment assays and ONMD. Figure 3 displays the results obtained by the two methods after a 40 min-long attachment period. These results confirm our working hypothesis that strains whose cells have a low attachment capacity (located below the 50% adhesion line) have cells that display more freedom in their displacements (located above the blue dashed horizontal line). Inversely, *C. albicans* strains that show strong cell binding (located above the brown dashed line) display a smaller ONMD (are situated below the blue dashed line of the upper graph).

To highlight a putative correlation between classical adhesion tests and ONMD measurements, we displayed both data sets onto the same graph and fitted it with a first order polynomial as depicted in Figure 4. The R-square of the linear fit is 0.77, which indicates a high correlation.

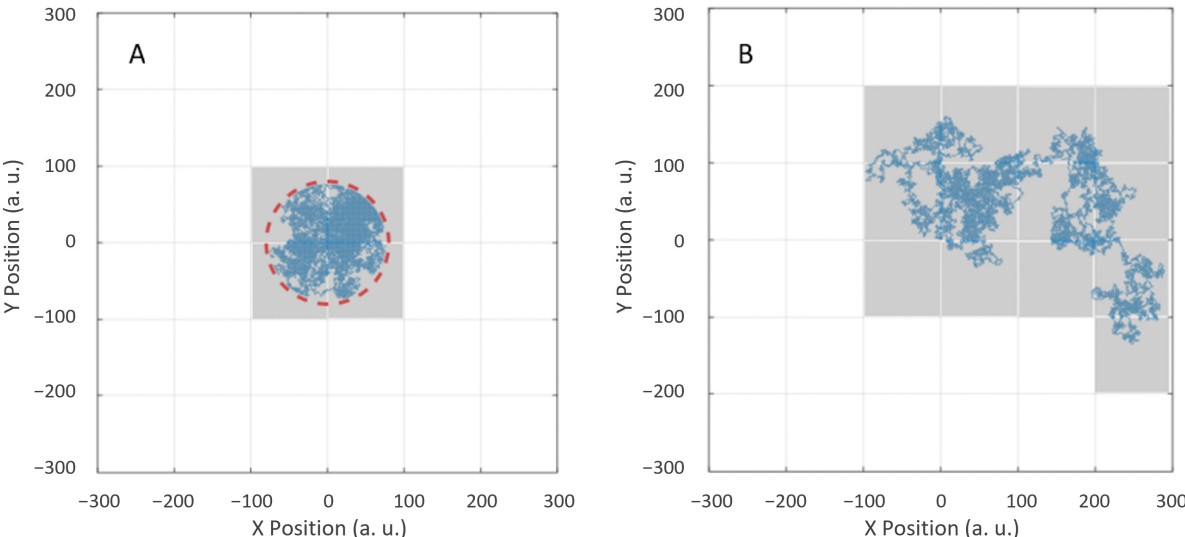

**Figure 2.** Random walk simulations of a constrained (**A**) and non-constrained (**B**) particle during 50,000 time steps. The "rope" that constrained the particle in frame A. had a length of 80 a.u. and is attached at the center of the graph. The red circle represents the maximum distance to which the particle can move if constrained by the rope. The number of pixels (gray squares) visited by the particle is higher in the non-constrained case ($n = 13$) than in the constrained one ($n = 4$).

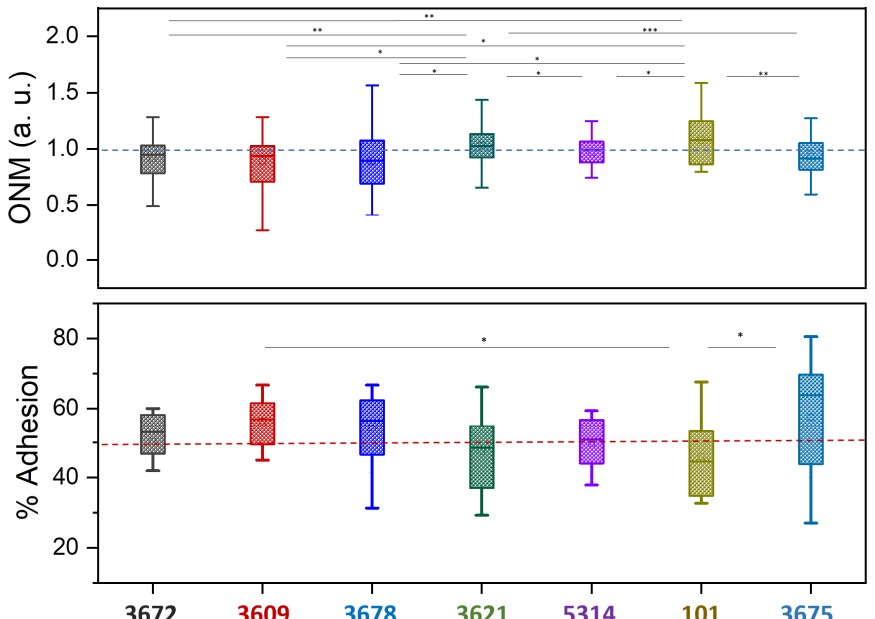

**Figure 3.** Adhesion results for 7 different *C. albicans* strains. The lower panel represents traditional adhesion test results, whereas the upper panel shows the corresponding ONMD data. The upper horizontal blue dashed line indicates the average displacement value of the cells at the beginning of the ONMD experiments (t = 0 min). The lower red dashed horizontal line indicates that 50% of the cells are attached to the fibronectin-coated surface in the adhesion tests. * $p < 0.05$; ** $p < 0.01$; *** $p < 0.001$. Number of cells per strain measured in ONM experiments: 3672 = 37, 3609 = 28, 3678 = 28, 3621 = 117, 5314 = 52, 101 = 28 and 3675 = 60. Number of replicates per strain measured in adhesion experiments: 3672 = 10, 3609 = 12, 3678 = 11, 3621 = 7, 5314 = 9, 101 = 11 and 3675 = 19.

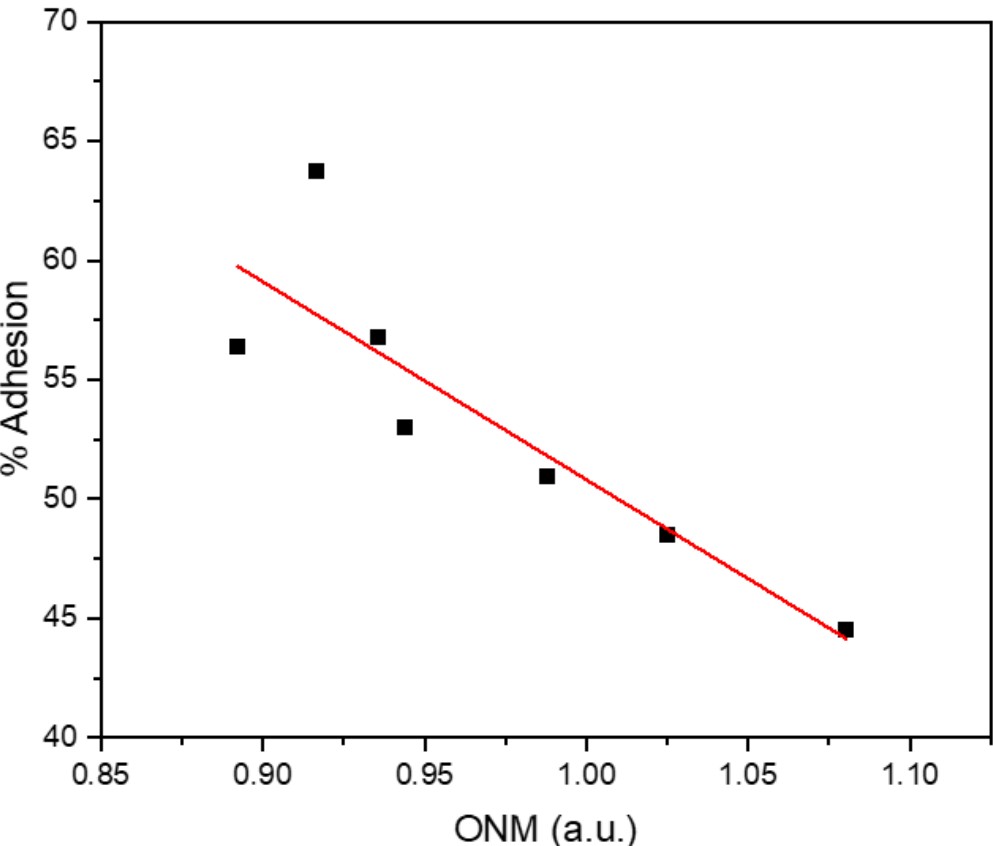

**Figure 4.** Linear fit of the traditional (percentage of adhesion) vs. the ONMD adhesion measurements results. The fitting equation has a slope-intercept form (y = a + bx), with intercept (a): 133.72 $\pm$ 19.88 and slope (b): $-82.90 \pm 20.47$.

As mentioned previously, ONMD permits to monitor the adhesion force as a function of time. Figure 5 shows the evolution of the nanomotion of different *C. albicans* strains during 40 min. As visible on this graph, the different strains behave differently upon exposure to fibronectin-coated surfaces. Certain strains increase their adhesion (reduction of the optical nanomotion signal) as a function of time, such as strains CEC3672, CEC3609, and CEC3678, whereas others, such as CEC101, reduce adhesion, and some, CEC3621, CEC5314, and CEC3675, show hybrid behavior during the measurement period. This type of information is relatively easily accessible by ONMD. It provides valuable information about yeast attachment dynamics and subtle differences in the attachment processes of the different strains.

The free and constrained random walk simulations demonstrated that free cells cover a larger distance than attached ones. To confirm this behavior with attached living cells, we displayed the displacement trajectories of a strongly (3672) and a poorly attached (101) cell. Figure 6 displays the trajectories of the two cells during 200 frames (i.e., about 6.7 s).

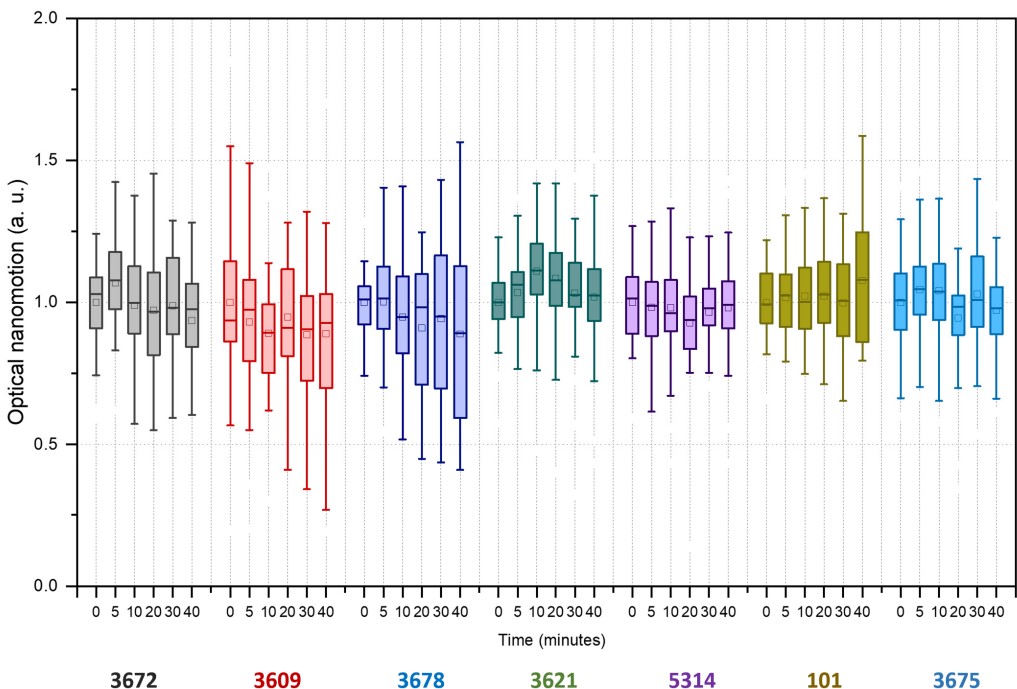

**Figure 5.** Evolution of the nanomotion of different *C. albicans* strains as a function of time. As shown, the different strains behave differently upon exposure to fibronectin-coated surfaces. The adhesion values for each strain are normalized to time 0. Number of cells: 3672: 0 min = 27, 5 min = 32, 10 min = 31, 20 min = 35, 30 min = 31 and 40 min = 32; 3609: 0 min = 35, 5 min = 37, 10 min = 37, 20 min = 33, 30 min = 27 and 40 min = 27; 3678: 0 min = 24, 5 min = 27, 10 min = 23, 20 min = 25, 30 min = 24 and 40 min = 24; 3621: 0 min = 54, 5 min = 59, 10 min = 69, 20 min = 74, 30 min = 71 and 40 min = 71; 5314: 0 min = 42, 5 min = 60, 10 min = 61, 20 min = 36, 30 min = 25 and 40 min = 35; 101: 0 min = 45, 5 min = 52, 10 min = 50, 20 min = 46, 30 min = 31 and 40 min = 28; 3675: 0 min = 41, 5 min = 42, 10 min = 48, 20 min = 37, 30 min = 35 and 40 min = 25.

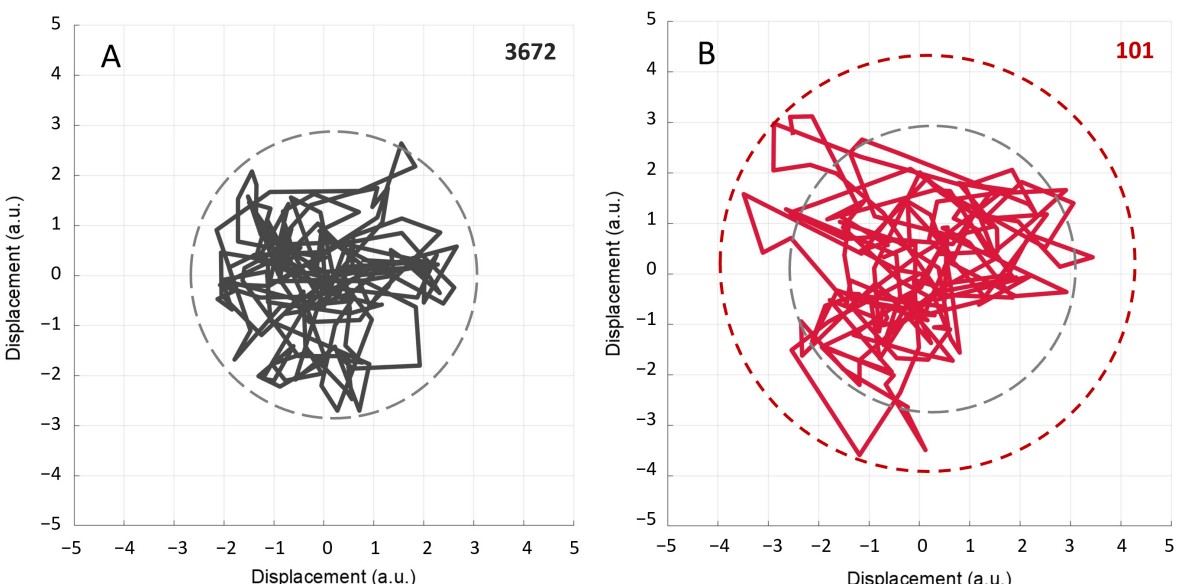

**Figure 6.** Typical path of single (**A**) CEC3672 and (**B**) 101 *C. albicans* cells. As depicted in this figure, the lower adherent cell (101- shown in red color) moves to larger distances than the higher adherent one (CEC3672- shown in gray color). The dashed circles correspond to the maximal distance reached by both cell types. In (**B**), both embedding circles are displayed for comparison.

## 4. Discussion

According to our experiments, ONMD can be used to determine the adhesion of yeast cells onto flat surfaces in a very rapid, cost-effective, and experimentally simple way. Recording by classical optical microscopy the nanomotion of living cells and processing the movies with a dedicated software permits not only to determine the adhesion, but also follow its modifications as a function of time. The technique is very straightforward and does not require any force-applying devices. The living cells' spontaneous nanomotion serves as a force generator, and the cellular attachment to the substrate constrains cellular displacements. Importantly, the measurement does not break the cell–substrate bond, and therefore enables measuring the adhesion as a function of time or as a function of various chemicals added in the analysis chamber at a given experimental time point. The evolution of the adhesion as a function of time is a poorly known parameter that is relatively difficult to measure with classical adhesion tests. It most likely reflects the speed at which different adhesive molecules that are present on the cell wall bind to the substrate, and could be a pertinent parameter to identify a given strain or inform about the invasive potential of the studied strain. These last two hypotheses would need additional experiments to be confirmed or invalidated.

The proposed technique has numerous advantages compared to traditional adhesion tests: it is extremely rapid, does not require any force-applying device, does not destroy the cell–substrate bond during the measurement, and can provide information on the adhesion evolution as a function of time. The experimental setup is also very simple and is limited to a traditional optical microscope equipped with a camera and dedicated image-processing software. Among the disadvantages, we can cite the difficulty, for the moment, of extracting absolute adhesion force values limiting the technique to relative measurements. Additional fluid dynamic studies, considering the geometry of the studied cell in addition to other physical parameters, could lead to some models that might correlate absolute adhesion values to nanomotion. A possible alternative could be calibration by an absolute adhesion measurement such as AFM, and its correlation with the nanomotion values.

Another limitation of the technique is the difficulty to separate the nanomotion itself from adhesion. Nanomotion measurements on cell-repellant surfaces could serve as a calibration technique to which the nanomotion on adherent surfaces could be normalized.

## 5. Conclusions

We introduced a new method to detect the dynamics of cellular adhesion. The new method is based on observing the cellular nanomotion of single cells by optical microscopy during the adhesion process. As a proof of concept, we successfully demonstrated the dynamics of *C. albicans*' adhesion to a fibronectin-coated surface. Among the advantages of the technique, we can mention its rapidity, the simplicity of the setup and the low cost of the required instrumentation. The technique is also cost effective, since the very same cell type can be employed in the very same experimental conditions to compare different adhesive surfaces. Importantly, the method offers the possibility of monitoring the evolution of the adhesion as a function of time without the need to destroy the link between the living organism and its substrate.

*C. albicans* is known to become a pathogen in patients having an impaired immune system. Its ability to adhere to catheter surfaces can be considered as a fundamental virulence feature [50]. The proposed technique could be very useful to rapidly evaluate *C. albicans*' adhesive properties on different medical devices. We are convinced that the technique can also be applied to bacterial adhesion too. In this last case, it would dramatically increase its application spectrum to fields such as medical material development, or antibacterial surface treatments.

**Author Contributions:** Conceptualization, S.K. and M.I.V.; methodology, S.K., M.I.V. and R.G.W.; software, S.K.; validation, M.I.V., S.L.-L., M.-E.B., C.d. and R.G.W.; investigation, S.K. and M.I.V.; resources, S.L.-L., M.-E.B., C.d., R.G.W. and S.K.; data curation, M.I.V.; writing—original draft prepa-

ration, S.K.; writing—review and editing, M.I.V., S.L.-L., M.-E.B., C.d., R.G.W. and S.K.; visualization, S.K.; supervision, S.K., R.G.W., S.L.-L. and C.d.; project administration, S.K.; funding acquisition, S.L.-L., M.-E.B., C.d., R.G.W. and S.K. All authors have read and agreed to the published version of the manuscript.

**Funding:** This research was funded by the Belgian Federal Science Policy Office (Belspo) and the European Space Agency, grant number PRODEX project Flumias Nanomotion; The Research Foundation—Flanders (FWO), grant numbers AUGE/13/19 and I002620; FWO-SNSF, grant number 310030L_197946, M.I.V., and S.K., S.L.-L. and C.d. were supported by Swiss National Science Foundation (SNSF) grant CRSII5_173863. Work in the laboratory of C.d. is supported by the Agence Nationale de Recherche (ANR-10-LABX-62-IBEID).

**Institutional Review Board Statement:** Not applicable.

**Informed Consent Statement:** Not applicable.

**Data Availability Statement:** The data presented in this study are available on request from the corresponding author.

**Acknowledgments:** The authors thank R. Horváth and I. Székács for highly constructive discussions.

**Conflicts of Interest:** The authors declare no conflict of interest.

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
