# Peer review of "Candida albicans Adhesion Measured by Optical Nanomotion Detection"

_fermentation, doi:10.3390/fermentation9110991_

Round 1

Reviewer 1 Report

Comments and Suggestions for Authors

Revie: Villalba et al.

The authors present a propose a novel, nanomotion-based, technique to study cellular adhesion.

For this they use Candida albicans as a model and aim at simplifying cell-substrate adhesion.

Qs:

1.       The authors use fibronectin-coated glass slides. What about con-A-coated glass slides? These are often used in yeast microscopy to immobilize cells.

2.       The use of different C. albicans strains requires some explanation. Mutants in cell surface proteins should be available in the Candida community.

3.       The YPD formula is not standard; why so?

4.       Figure 1 - the cells in 3a and 3b are not immersed, right? Here one only looks at CFUs. Thus the image is misleading. Also, three cells are not yet a colony…

5.       On the evolution of adhesion over time: In S. cerevisiae adhesion, that is cell-cell adhesion, develops over time during nutrient limitation. This should be discussed.

6.       Nanomotion vs adhesion: maybe con-A results in better adhesion of cells to glass slides?

7.       Imaging may result in heating the sample via illumination. How was this handled? E.g. via LED illumination vs halogen lamp light?

Author Response

Reviewer 1

The authors present a propose a novel, nanomotion-based, technique to study cellular adhesion.

For this they use Candida albicans as a model and aim at simplifying cell-substrate adhesion.

Qs:

  1. The authors use fibronectin-coated glass slides. What about con-A-coated glass slides? These are often used in yeast microscopy to immobilize cells.

We focused in this manuscript to the adhesion to fibronectin since it is a physiologically relevant molecule that is present in patients basal lamina and that plays a role in Candida adhesion in immunocompromised patients. It has been shown that fibronectin is an important protein ligand of the host extracellular matrix (ECM) that plays an essential role in C. albicans adhesion (1). Furthermore, targeting fibronectin has shown to alter C. albicans biofilm formation (2). As is demonstrated in this manuscript, a nanomotion-based adhesion test could be successfully developed by selecting fibronectin as the adhesion ligand.

1-Calderone, R.A.; Scheld, W.M. Role of fibronectin in the pathogenesis of candidal infections. Rev. Infect. Dis. 1987, 9, S400–S403

2-Nett, J.E.; Cabezas-Olcoz, J.; Marchillo, K.; Mosher, D.F.; Andes, D.R. Targeting fibronectin to disrupt in vivo Candida albicans biofilms. Antimicrob. Agents Chemother. 2016, 60, 3152–3155.

  1. The use of different C. albicans strains requires some explanation. Mutants in cell surface proteins should be available in the Candida community.

We used different C. albicans strains since they possess different adhesive properties to assess the validity of the method. The Candida strains we used in this manuscript are available upon request from C. D’Enfert. The references for the strains are indicated in the Materials Methods section 2.1, meaning that they are available in the Candida community.

  1. The YPD formula is not standard; why so?

The reviewer is right, there was an error in the YPD formula. We use the following formula : D-glucose (Gibco™ 15023021) 20 gr/l, Peptone (Millipore 82303) 20 g/l, Yeast extract 10 g/l, agar (plates only) 20g/l diluted in distilled water. This has now been corrected in the manuscript.

  1. Figure 1 - the cells in 3a and 3b are not immersed, right? Here one only looks at CFUs. Thus the image is misleading. Also, three cells are not yet a colony…

The cells in 3a and 3b are immersed in YPD and SDA agar as specified in the figure caption and the Materials and Methods section. Yes, cells in 3a and 3b are immersed in agar. To clarify this point we modified the figure caption as follows: “…deposited into a new Petri dish with YPD agar (3a). The single cells eventually divide and grow into colonies (4a) that permits to estimate their number.  …. » The growing colony in the figure is represented as a cross-section of the colony. In the figure it is represented as a cross-section through 3 cells.

  1. On the evolution of adhesion over time: In S. cerevisiae adhesion, that is cell-cell adhesion, develops over time during nutrient limitation. This should be discussed.

We only studied single cell adhesion to fibronectin over a relatively short time frame (40 min). In this case, they are no nutrient limitations. This situation is totally different from cell-cell interaction that can lead to yeast flocs (3)

3- Goossens KV, Ielasi FS, Nookaew I, Stals I, Alonso-Sarduy L, Daenen L, Van Mulders SE, Stassen C, van Eijsden RG, Siewers V, Delvaux FR, Kasas S, Nielsen J, Devreese B, Willaert RG. Molecular mechanism of flocculation self-recognition in yeast and its role in mating and survival. mBio. 2015 Apr 14;6(2):e00427-15. doi: 10.1128/mBio.00427-15. PMID: 25873380; PMCID: PMC4453552.

  1. Nanomotion vs adhesion: maybe con-A results in better adhesion of cells to glass slides?

As elaborated in our answer to question one, studying adhesion of C. albicans is more physiological relevant than the adhesion to the plant lectin concanavalin A.

  1. Imaging may result in heating the sample via illumination. How was this handled? E.g. via LED illumination vs halogen lamp light?

We are convinced that heating of the sample by the illumination has no significant influence on the measurements since the cells were illuminated only during 10 s to record the nanomotion movies. Recordings were performed after 5, 10, 20, 30 and 40 min at room temperature (Materials and Methods) keeping the analysis chamber temperature at a constant value.

We added the following sentence to the Materials and Methods section “2.4 Video recording and data processing”: Nanomotion movies were recorded at time 0, 5, 10, 20, 30 and 40 min. »

Reviewer 2 Report

Comments and Suggestions for Authors

The manuscript “Candida albicans adhesion measured by optical nanomotion detection” present a novel technique for measuring adherence of C. albicans cells on a fibronectin-coated glass surface. Through this technique, called optical nanomotion detection (ONMD), authors examined the nanomotion amplitude of seven different C. albicans strains (CEC 3672, CEC 3609, CEC 3678, CEC 3621, SC 5314, 101 and CEC 3675) deposited on optical quality petri dish coated with fibronectin. They measured the amplitude of nanomotion of these strains and compared it to the values obtained by a classical adhesion test. The work presented is novel, since the authors have developed a technique that allows the measurement of cell adhesion at different times using a simple method that does not require expensive equipment. Despite this, I believe that the work is unpublishable in its current form. The following considerations must be taken into account prior to consider this work for publication:

(1)  Why do you use fibronectin? Is it proven that C. albicans binds to it?

(2)  With respect to the adhesion test

-          Is it an assay that you have developed and use to measure adherence, or is it widely used? This should be explained in the text. If there is already a protocol for this assay, add a reference.

-          Add the trademark and supplier of the 6-well plates used for this assay.

-          After adding the fibronectin and incubating for one hour, you say you remove the liquid, but don't you also wash the wells with PBS?

-          After this step, you say you add 250 cells resuspended in YPD: How do you count these cells? using Neubaeur chamber? why do you use this number of cells?

-          Once you add the fungal cells, incubate for 25 minutes, at what incubation temperature? with or without shaking? add this data in the protocol.

-          In the protocol you say that after this time of incubation, you remove the liquid (supernatant containing the unattached cells) and the attached cells remain on the plate. Don't you wash the cells that remain on the plate with PBS to remove any unattached cells? How do you know if the supernatant that you remove really contains all the unattached cells and that the ones that remain on the plate are really the attached ones?

-          Why do you use SDA for the cells that remain attached and YPD for the unattached cells?

-          Why do yeasts incubate for 48 h?

(3)  Regarding the measurement method (ONMD)that you present in this work, I believe that since it is the most important part of it, it should be better explained, since the way it is presented is not very clear. Better explain the software developed to measure nanomotion, what is measured?, what data is obtained?. Explain well if all the cells present in the culture are measured or only some of them are selected? If only some are selected, how is this selection carried out? How many are selected? In the materials and methods part, when this technique is explained, reference number 6 should be cited, or those that you consider relevant, since this technique had already been developed and in this work the novelty is that it is applied to measure adhesion.

-          The petri dishes used are glass? add this information because it is important, since it is the surface on which you are testing adhesion. The 6-well plates used in the adhesion method should be of the same material, since you then compare the adhesion process by both techniques.

-          On line 109 it says that the petri dishes used are low uncoated, what does it mean?

-          Same as for the previous technique, after incubation with fibronectin, do you not wash the plate with PBS?

-          At this concentration used, can the cells be visualized correctly by microscopy, since in order to measure the individual nanomotion of each yeast, they should not be overlapped or too close together?

-          Better explain the part “2.3 Constrained random walk simulation

-          I would put what is presented under the title "Video recording and data processing" higher up, and without a title, so that the technique used is better explained.

-          In figure 1 it says that the adhesion technique consists in depositing yeast cells onto a fibronectin-coat Petri dish for 40 min but in the explanation of the methodology it says that it is for 25 min

(4)  In the results part, you say "To confirm our hypothesis according to which a low adhesion force corresponds to a higher displacement freedom, we carried on constrained and non-constrained random walk simulations”. To do this simulation, did you take experimental data? you only did it with a single cell? Figure 2 says that it is based on a single particle, but it seems to me that the behavior should be analyzed for many particles to get closer to the global behavior at the population level. In addition to this, in the graphs presented in figure 2 the measurement units should be added to the x and y axes. What does 50000 times steps mean? and 80 au?

(5)  I think that the most convenient way to present the results obtained for the adhesion tests using the two techniques is using a box plot or a violin plot. In figure 3 two different graphs are used, this should be homogenized. In the legend of figure 3 it says: “The upper panel represents traditional adhesion test results whereas the lower panel shows the corresponding ONMD data” which is wrong since they are presented in the image inverted. Also, the line is red instead of brown as mentioned in the legend. Also in this figure it is necessary to clarify from which number of cells measured these data are obtained. In the case of the values obtained by the ONMD technique, what does a.u. mean? should be clarified.

(6)  The results shown in Figure 5 are not well explained in the text.

(7)  Why is Figure 6 showing the data obtained for a single cell or not the average of a population of cells?

(8)  As you mention in the discussion part: “Another limitation of the technique is the difficulty to separate the nanomotion itself from adhesion. Nanomotion measurements on cell repellant surfaces could serve as calibration to which the nanomotion on adherent surfaces could be normalized”. I believe that it is essential to have control of cells that do not adhere and measure their nanomotion, and compare these values with those obtained for the cells that are adhered. Is it feasible to use a Petri dish without prior treatment with fibronectin, and evaluate cell mobility under these conditions? and use this treatment as a control for non-adherent cells.

Author Response

The manuscript “Candida albicans adhesion measured by optical nanomotion detection” present a novel technique for measuring adherence of C. albicans cells on a fibronectin-coated glass surface. Through this technique, called optical nanomotion detection (ONMD), authors examined the nanomotion amplitude of seven different C. albicans strains (CEC 3672, CEC 3609, CEC 3678, CEC 3621, SC 5314, 101 and CEC 3675) deposited on optical quality petri dish coated with fibronectin. They measured the amplitude of nanomotion of these strains and compared it to the values obtained by a classical adhesion test. The work presented is novel, since the authors have developed a technique that allows the measurement of cell adhesion at different times using a simple method that does not require expensive equipment. Despite this, I believe that the work is unpublishable in its current form. The following considerations must be taken into account prior to consider this work for publication:

(1)  Why do you use fibronectin? Is it proven that C. albicans binds to it?

We focused in this manuscript to the adhesion to fibronectin since it is a physiologically relevant molecule that is present in patients basal lamina and that plays a role in Candida sp adhesion in immunocompromised patients. It has been shown that fibronectin is an important protein ligand of the host extracellular matrix (ECM) that plays an essential role in C. albicans adhesion (1). Furthermore, targeting fibronectin has shown to alter C. albicans biofilm formation (2). As is demonstrated in this manuscript, a nanomotion-based adhesion test could be successfully developed by selecting fibronectin as the adhesion ligand.

1- Calderone, R.A.; Scheld, W.M. Role of fibronectin in the pathogenesis of candidal infections. Rev. Infect. Dis. 1987, 9, S400–S403.

2- Nett, J.E.; Cabezas-Olcoz, J.; Marchillo, K.; Mosher, D.F.; Andes, D.R. Targeting fibronectin to disrupt in vivo Candida albicans biofilms. Antimicrob. Agents Chemother. 2016, 60, 3152–3155.

(2)  With respect to the adhesion test

 Is it an assay that you have developed and use to measure adherence, or is it widely used? This should be explained in the text. If there is already a protocol for this assay, add a reference.

This adhesion test was conducted according to the protocol previously described (3-5) with a few modifications: the cell layer was replaced with fibronectin in this study. After that, we followed the protocol exactly as described in the citations.

3- Murciano, C.; Moyes, D.L.; Runglall, M.; Tobouti, P.; Islam, A.; Hoyer, L.L.; Naglik, J.R. Evaluation of the Role of Candida albicans Agglutinin-Like Sequence (Als) Proteins in Human Oral Epithelial Cell Interactions. PLoS ONE 2012, 7, e33362.

4- Schönherr, F.A.; Sparber, F.; Kirchner, F.R.; Guiducci, E.; Trautwein-Weidner, K.; Gladiator, A.; Sertour, N.; Hetzel, U.; Le, G.T.T.; Pavelka, N.; et al. The intraspecies diversity of C. albicans triggers qualitatively and temporally distinct host responses that determine the balance between commensalism and pathogenicity. Mucosal Immunol. 2017, 10, 1335–1350.

5- Anne-Céline Kohler, Leonardo Venturelli, Abhilash Kannan, Dominique Sanglard, Giovanni Dietler, Ronnie Willaert and Sandor Kasas. Yeast Nanometric Scale Oscillations Highlights Fibronectin Induced Changes in C. Albicans. Fermentation 2020, 6(1), 28.

-          Add the trademark and supplier of the 6-well plates used for this assay.

The data has now been added to the manuscript.

-          After adding the fibronectin and incubating for one hour, you say you remove the liquid, but don't you also wash the wells with PBS?

Following incubation with fibronectin, the functionalized surface was washed with PBS. We have now added this data to the manuscript.

-          After this step, you say you add 250 cells resuspended in YPD: How do you count these cells? using Neubaeur chamber? why do you use this number of cells?

Following the measurement of the optical density, we adjusted the number of cells by assuming that one OD600 corresponds to approximately 3×10^7 cells/mL (6).

6- Short protocols in molecular biology, Fred Ausubel et al., 5th ed. Vol. 2 Wiley, New York. pp. 13-9

-          Once you add the fungal cells, incubate for 25 minutes, at what incubation temperature? with or without shaking? add this data in the protocol.

Following the addition of the fungal cells, the incubation was conducted at 30°C without shaking. This data is now added in the manuscript.

-          In the protocol you say that after this time of incubation, you remove the liquid (supernatant containing the unattached cells) and the attached cells remain on the plate. Don't you wash the cells that remain on the plate with PBS to remove any unattached cells? How do you know if the supernatant that you remove really contains all the unattached cells and that the ones that remain on the plate are really the attached ones?

Pipetting the liquid up and down, while keeping the multiwell plate tilted, is used to remove all non-adherent cells from the multiwell plate. The liquid was then inoculated onto a new agar plate. To remove all unattached cells, a small volume of fresh medium is used instead of PBS for washing. This liquid is also inoculated on the previous agar plate.

-          Why do you use SDA for the cells that remain attached and YPD for the unattached cells?

We used SDA and YPD according to the protocol that has already been published (7-9). Since SDA is more transparent than YPD, it provides a better count of adhered cells at the bottom of the petri dish

7- Murciano, C.; Moyes, D.L.; Runglall, M.; Tobouti, P.; Islam, A.; Hoyer, L.L.; Naglik, J.R. Evaluation of the Role of Candida albicans Agglutinin-Like Sequence (Als) Proteins in Human Oral Epithelial Cell Interactions. PLoS ONE 2012, 7, e33362.

8- Schönherr, F.A.; Sparber, F.; Kirchner, F.R.; Guiducci, E.; Trautwein-Weidner, K.; Gladiator, A.; Sertour, N.; Hetzel, U.; Le, G.T.T.; Pavelka, N.; et al. The intraspecies diversity of C. albicans triggers qualitatively and temporally distinct host responses that determine the balance between commensalism and pathogenicity. Mucosal Immunol. 2017, 10, 1335–1350.

9- Anne-Céline Kohler, Leonardo Venturelli, Abhilash Kannan, Dominique Sanglard, Giovanni Dietler, Ronnie Willaert and Sandor Kasas. Yeast Nanometric Scale Oscillations Highlights Fibronectin Induced Changes in C. Albicans. Fermentation 2020, 6(1), 28. DOI: https://doi.org/10.3390/fermentation6010028

-          Why do yeasts incubate for 48 h?

Yeast cells were incubated for 48 hours to facilitate the counting of CFU. The size of the colonies after 48h allows the counting of the colonies to be performed more easily than by incubating for 24 hours.

(3)  Regarding the measurement method (ONMD)that you present in this work, I believe that since it is the most important part of it, it should be better explained, since the way it is presented is not very clear. Better explain the software developed to measure nanomotion, what is measured? what data is obtained?

As suggested, we splitted section 2.2 into « Adhesion assay » and « 2.3 Optical Nanomotion adhesion assay ». We also merged the former section « 2.5 Video recording and data processing » with the new section 2.3 : « 

2.2 Adhesion assay

Two different methods were used to measure C. albicans adhesion as illustrated in Figure 1. The first method (16-18) consisted of incubating freely floating yeast cells in wells coated with fibronectin. A 6-well culture plate (add type, manufacturer) was incubated for an hour with fibronectin (50 µg/ml) (fibronectin human plasma, lyophilized powder, BioReagent, F2006). The liquid was gently removed, and 250 cells suspended in 500 µl of YPD medium were added to each well. After 25 minutes, the supernatant containing the freely floating cells was removed (Figure1, 2a) and inoculated in a new YPD agar Petri dish (3a); attached cells were immersed in Sabourand Dextrose (SDA) agar (3b), by adding melted agar at 40°C. SDA was composed of d-glucose (Gibco™ 15023021) 40 gr/l, peptone (Millipore 82303) 10 g/l, agar 15g/ l diluted in distilled water. After an incubation of 48 h the yeast cells in both Petri dishes formed colonies (4a, 4b). The % adhesion was calculated by dividing the number of grown cells in SDA agar by the total number of grown cells (in both YPD and SDA agar) for each strain.

Figure 1. Two different techniques are used to measure adhesion: 1-4 Classical technique based on the ratio between floating and adhering cells and 5. nanomotion-based measurement. The classical method (1-4b) consists in depositing yeast cells onto a fibronectin-coat Petri dish for 40 min (1). The supernatant is than removed (2a), deposited into a new Petri dish with YPD agar (3a). The single cells eventually divide and grow into colonies (4a) that permits to estimate their number. The cells that remained attached to the fibronectin covered Petri dish (2b) were also covered with agar (3b) and developed CFUs too (4b). The nanomotion-based measurements (5) were performed by recording movies of 10 s.

2.3 Optical nanomotion adhesion method

ONMD was performed by recording 10 s long movies with an inverted optical microscope (Zeiss Observer Z.1) of the attachment of C. albicans cells. We used a 63x oil immersion objective, and a PCO Edge 5.5 camera. Petri dishes (µ-Dish 35 mm, low, uncoated, 80131 IBIDI) were incubated for an hour with fibronectin (50 µg/ml) (BioReagent, F2006) according to the protocol recommended at www.ibidi.com. The liquid was removed and 1.5 ml of YPD liquid medium with 20 µl of overnight culture was added to the Petri dish with the fibronectin layer. Two minutes after placing the plates under the microscope, the first video (time 0) was recorded, followed by videos at 5, 10, 20, 30, and 40 min (Figure1 (5)). All the measurements were carried out at room temperature without phase contrast nor fluorescent staining. “

- Explain well if all the cells present in the culture are measured or only some of them are selected? If only some are selected, how is this selection carried out? How many are selected?

Only individual cells (those that were not in contact with other cells) were selected. We added the number of cells that were used to calculate the average to the figure caption. We selected al least 20 cells to have a statistically significant number of cells per condition.

Number of cells in Fig 5: 3672 : 0 min=27, 5 min=32, 10min=31, 20 min=35, 30 min=31 and 40min=32. 3609 : 0 min=35, 5 min=37, 10 min=37, 20 min=33, 30 min=27 and 40 min=27. 3678 : 0 min=24, 5 min=27, 10 min=23, 20 min=25, 30 min=24 and 40 min=24. 3621 : 0 min=54, 5 min= 59, 10 min=69, 20 min=74, 30 min=71 and 40 min=71. 5314: 0 min=42, 5 min=60, 10 min=61, 20 min=36, 30 min=25 and 40 min=35. 101 : 0 min=45, 5 min=52, 10 min=50, 20 min=46, 30 min=31 and 40 min=28. 3675: 0 min=41, 5 min=42, 10 min=48, 20 min=37, 30 min=35 and 40 min=25.

- In the materials and methods part, when this technique is explained, reference number 6 should be cited, or those that you consider relevant, since this technique had already been developed and in this work the novelty is that it is applied to measure adhesion.

The reference [6] was present in the Materials and Methods section 2.5 ; in section 2.3 in the revised manuscript.

-The petri dishes used are glass? add this information because it is important, since it is the surface on which you are testing adhesion. The 6-well plates used in the adhesion method should be of the same material, since you then compare the adhesion process by both techniques.

Petri dishes with a fibronectin coated polymer coverslip bottom were used for the nanomotion experiments. This information is now added to section 2.4 of Materials and Methods : « The nanomotion-based measurement (5) consisted in depositing the yeast cells onto a fibronectin coated Petri dish (with optical bottom) (µ-Dish 35 mm, low, uncoated, 80131 IBIDI) and in recording their nanometer scale total displacements (nanomotion) during 10 s [6]. »

The fibronectin functionalization is sepecified in the lines 101 and 102 of the section 2.2: Adhesion assay and the lines 12-128 of the section 2.3: Optical nanomotion adhesion method

 - On line 109 it says that the petri dishes used are low uncoated, what does it mean?

A comma was missing between low and uncoated, meaning that we used a Petri dish with a low walls as specified by the manufacturer (https://ibidi.com/dishes/14--dish-35-mm-low-ibitreat.html#/26-surface_modification-uncoated_15_polymer_coverslip_hydrophobic_sterilized/30-pcs_box-60_individually_packed)

- Same as for the previous technique, after incubation with fibronectin, do you not wash the plate with PBS?

Yes, a wash with PBS was performed according to the instructions provided in the IBIDI functionalization protocol. The manuscript now contains this information.

-  At this concentration used, can the cells be visualized correctly by microscopy, since in order to measure the individual nanomotion of each yeast, they should not be overlapped or too close together?

Yes, we could observe the individual cells. We did not analyse grouped cells.

- Better explain the part “2.3 Constrained random walk simulation”

We added a reference and better specified the method : « A fixed step length random walk (19) was simulated with a Matlab (R2023a) computer program. We simulated a massless particle submitted to a random walk during 50’000 time cycles. »

- I would put what is presented under the title "Video recording and data processing" higher up, and without a title, so that the technique used is better explained.

We adapted this as suggested by the reviewer.

- In figure 1 it says that the adhesion technique consists in depositing yeast cells onto a fibronectin-coat Petri dish for 40 min but in the explanation of the methodology it says that it is for 25 min.

The incubation time was 25 minutes. It is now corrected in the caption figure.

(4)  In the results part, you say "To confirm our hypothesis according to which a low adhesion force corresponds to a higher displacement freedom, we carried on constrained and non-constrained random walk simulations”. To do this simulation, did you take experimental data? you only did it with a single cell? Figure 2 says that it is based on a single particle, but it seems to me that the behavior should be analyzed for many particles to get closer to the global behavior at the population level. In addition to this, in the graphs presented in figure 2 the measurement units should be added to the x and y axes. What does 50000 times steps mean? and 80 au?

This simulation is based on a massless particle that is constrained and unconstrained. This simulation is intended to investigate the relation between adhesion force and displacement freedom. "50000 time steps" is the number of time cycles for a particle, which is the duration of a random walk.  80 a.u. is an abbreviation for 80 arbitrary length units, which means the length of the ropes that limited the distance at which particles could diffuse. The measure units (a.u.) were added in the graphs.

(5)  I think that the most convenient way to present the results obtained for the adhesion tests using the two techniques is using a box plot or a violin plot. In figure 3 two different graphs are used, this should be homogenized. In the legend of figure 3 it says: “The upper panel represents traditional adhesion test results whereas the lower panel shows the corresponding ONMD data” which is wrong since they are presented in the image inverted. Also, the line is red instead of brown as mentioned in the legend. Also, in this figure it is necessary to clarify from which number of cells measured these data are obtained. In the case of the values obtained by the ONMD technique, what does a.u. mean? should be clarified.

ONM results are now represented as boxplots. The number of cells and replicates per strain measured in ONM and adhesion assays are now added in the manuscript (figure 3 caption). Number of cells per strain measured in ONM experiments: 3672= 37, 3609= 28, 3678= 28, 3621= 117, 5314= 52, 101= 28 and 3675= 60. Number of replicates per strain measured in adhesion experiments: 3672= 10, 3609=12, 3678= 11, 3621= 7, 5314= 9, 101= 11 and 3675= 19. We corrected in the figure caption « lower » and « upper » panels as well as the color of the line to « red ». « a.u. » means arbitrary units

(6)  The results shown in Figure 5 are not well explained in the text.

To better explain the results respresented in Figure 5, we adapted in this section the following sentence: « Certain strains increase their adhesion (reduction of the optical nanomotion signal) as a function of time such as strains CEC3672, CEC 3609, and CEC3678, whereas others such as CEC101, reduce it adhesion, whereas some, suac CEC3621, CEC5314, and CEC3675, showed hybrid behavior during the measurement period. »

The results are now represented as a boxplot chart.

(7)  Why is Figure 6 showing the data obtained for a single cell or not the average of a population of cells?

We selected only the displacement of 2 cells to illustrate the typical single cell nanomotion during 10 s. Adding more displacements of cells on the same graph would make it unreadable.

(8)  As you mention in the discussion part: “Another limitation of the technique is the difficulty to separate the nanomotion itself from adhesion. Nanomotion measurements on cell repellant surfaces could serve as calibration to which the nanomotion on adherent surfaces could be normalized”. I believe that it is essential to have control of cells that do not adhere and measure their nanomotion, and compare these values with those obtained for the cells that are adhered. Is it feasible to use a Petri dish without prior treatment with fibronectin, and evaluate cell mobility under these conditions? and use this treatment as a control for non-adherent cells.

Yes, but in the aim of the present work, this test is not relevant since cells could interact with non-coated surface (electrostatic interactions). Therefore, this test will also not be a good control. In addition, the interaction of Candida cells to fibronectin can change the nanomotion pattern as studied previously (10)

10 - Kohler, A.-C.; Venturelli, L.; Kannan, A.; Sanglard, D.; Dietler, G.; Willaert, R.; Kasas, S. Yeast Nanometric Scale Oscillations Highlights Fibronectin Induced Changes in C. albicans. Fermentation 2020, 6, 28. https://doi.org/10.3390/fermentation6010028.

Reviewer 3 Report

Comments and Suggestions for Authors

The authors investigated cellular nanomotiom using optical nanomotion detection (ONMD) which can be performed with a simple microscope and a dedicated software. They compared cellular adhesion of 7 strains of Candida albicans using ONMD and concluded that this method correlates well with existing yet cumbersome technique. The results are as expected: stronger the adhesion, smaller the motion. The following needs to be addressed before considering for publication.

  1. Distances traveled are all in arbitrary units. Since the analysis is considering subpixel movements, it would also be necessary to discuss nanomotion in absolute terms. 
  2. In this case various strains of Candida were analysed. How to consider motion of different forms that are significantly different in cellular size.
  3. The authors are concluding that it may be possible to apply this to study bacteria which they have already published elsewhere. It is better to rewrite this part to reflect literature accurately in introduction itself.
  4. Figure 3 shows blue dotted line to reflect initial state. It is surprising that all strains have same adhesion in Figue 5 to begin with. It is not clear if any normalization was performed. 
  5. Figure 3 caption should be revised to reflect the figure accurately. The upper panel is ONMD and lower one shows traditional adhesion results.

Author Response

Reviewer 3

The authors investigated cellular nanomotiom using optical nanomotion detection (ONMD) which can be performed with a simple microscope and a dedicated software. They compared cellular adhesion of 7 strains of Candida albicans using ONMD and concluded that this method correlates well with existing yet cumbersome technique. The results are as expected: stronger the adhesion, smaller the motion. The following needs to be addressed before considering for publication.

  1. Distances traveled are all in arbitrary units. Since the analysis is considering subpixel movements, it would also be necessary to discuss nanomotion in absolute terms.

In order to discuss and correlate nanomotion with absolute values, we analyzed a movie we created where we can determine the movement values of the individual cells in micrometers. As seen in Figure 1, 3 cells were analyzed in this video.

Figure 1 : Upper panel: Optical and ONM images of the analyzed movie. The bottom panel shows the path of the three cells.

By performing a correlation between ONM and the displacement values in µm (Figure 2), we were able to derive the following equation:

Displacement (μm) = 4.8008 ONM displacement + 12.139

This equation will vary depending on the parameters used to acquire the image, including the objective, magnification, resolution, number of pixels, etc.

Figure 2: Table and graph illustrating the relationship between ONM and displacement in absolute values (µm)

  1. In this case various strains of Candida were analysed. How to consider motion of different forms that are significantly different in cellular size.

It not clear to us to which forms the reviewer is refering to. In this study, we evaluated only the planktonic yeast form of Candida albicans adhesion to fibronectin.

  1. The authors are concluding that it may be possible to apply this to study bacteria which they have already published elsewhere. It is better to rewrite this part to reflect literature accurately in introduction itself.

Up to now, we did not apply nanomotion to study bacterial adhesion. We did study the nanomotion of bacteria in solution in the frame of antibiotic susceptibility testing, but never used the technique to study bacterial adhesion.

  1. Figure 3 shows blue dotted line to reflect initial state. It is surprising that all strains have same adhesion in Figue 5 to begin with. It is not clear if any normalization was performed. 

As the review points out, in Figure 5, the adhesion is normalized to time 0. The manuscript now provides a better explanation of this.

Figure 3 caption should be revised to reflect the figure accurately. The upper panel is ONMD and lower one shows traditional adhesion results.

We corrected this mistake in the revised manuscript.

Round 2

Reviewer 2 Report

Comments and Suggestions for Authors

The current version of the manuscript is much improved over the previous version. The work presents novelty, and in these conditions I believe it deserves to be published in the journal.